# Replicating Spectral Baseline for Unambiguous Frequency Locking in Resonant Sensors

**DOI:** 10.3390/s24072318

**Published:** 2024-04-05

**Authors:** Andi Setiono, Wilson Ombati Nyang’au, Erwin Peiner

**Affiliations:** 1Laboratory for Emerging Nanometrology (LENA), Institute of Semiconductor Technology (IHT), Technische Universität Braunschweig, 38106 Braunschweig, Germany; wombati@kebs.org (W.O.N.); e.peiner@tu-braunschweig.de (E.P.); 2Research Center for Photonics—National Research and Innovation Agency (BRIN), South Tangerang 15314, Indonesia; nelf001@brin.go.id; 3Department of Metrology, Kenya Bureau of Standards (KEBS), Nairobi 00200, Kenya

**Keywords:** electrothermal-piezoresistive-cantilever sensor, thermal parasitic coupling, resonant MEMS sensor, phase-locked-loop, replica of baseline spectra

## Abstract

Electrothermal piezoresistive resonant cantilever sensors have been fabricated with embedded actuating (heating resistor) and sensing (piezo resistors) parts, with the latter configured in a Wheatstone bridge circuit. Due to the close spacing between these two elements, a direct thermal parasitic effect on the resonant sensor during the actuating-sensing process leads to asymmetric amplitude and reversing phase spectral responses. Such a condition affects the precise determination of the cantilever’s resonant frequency, *f*_0_. Moreover, in the context of phase-locked loop-based (PLL) resonance tracking, a reversing phase spectral response hinders the resonance locking due to its ambiguity. In this work, a replica of the baseline spectral was applied to remove the thermal parasitic effect on the resonance spectra of the cantilever sensor, and its capability was simulated through mathematical analysis. This replica spectral was subtracted from the parasitized spectral using a particular calculation, resulting in optimized spectral responses. An assessment using cigarette smoke particles performed a desired spectral shifting into symmetrical amplitude shapes and monotonic phase transitions, subsequently allowing for real-time PLL-based frequency tracking.

## 1. Introduction

Sensing technology is an integral and vital part of human life, with applications ranging from lifestyle, healthcare, and fitness to manufacturing and daily routines. Micro-electro-mechanical system (MEMS) is one of the leading sensing technologies; it has greatly revolutionized the development and market for sensors by providing tiny, fast responding [1,2], and highly reliable devices at a low cost. MEMS resonance-based sensors are one of the growing areas due to their scalability (i.e., benefits in size) [3,4,5], low energy consumption [6], and high-volume fabrication. Moreover, with the recent integration of MEMS sensors into gas and humidity sensing systems [7], a transformative shift in environmental detection is now an inevitable reality. MEMS sensors continue to play a revolutionary and cutting-edge role in environmental monitoring by utilizing their remarkable sensitivity, size scaling, and miniaturization [8,9].

Thermally actuated piezoresistive MEMS microsystems are capable of measuring at a high resolution [10,11,12,13] due to their small size. Electrothermal-based propulsion systems benefit from low fabrication complexity and high responsivity [14]. Utilizing piezoresistors as the sensing parts leads to a high signal-to-noise ratio, high shrinking capability, and compatibility with electronic components on the market. However, thermally-induced parasitic factors inside the sensor are a disadvantage that should be overcome and eliminated. The said thermal parasitic effects generate a Fano resonance (asymmetric shape) and a reversing spectral response at the sensing element’s output. Considering phase-locked loop-based (PLL) measurements, a reversing section response is usually inhibitive. Therefore, this work aims to optimize the spectral response of the sensor by de-embedding the parasitic signal from the measured signal. Several techniques have been employed to address this parasitic feedthrough effect (FE), including: a phase-controlled oscillation technique for operating microresonators beyond the nonlinear regime with on-chip FE de-embedding [15], the application of a unique design of thin-film piezoelectric-on-silicon (TPoS) MEMS resonator with a fully differential configuration to eliminate parasitic feedthrough effects [16], and the design and performance analysis of a PLL charge-pump with very low feedthrough characteristics [17]. These proposed methods highlight the high level of complexity in the technical approaches and analyses required to tackle feedthrough issues, emphasizing the importance of careful design innovation and control techniques to enhance the performance and operational stability of microresonators and related circuits.

In this investigation, the parasitic signal is estimated by imitating the baseline of the parasitized spectral responses, herein called a replica of baseline spectral. By creating a replica of the baseline spectral, we can further develop a computational model that accurately simulates the expected baseline of the sensor in the presence of parasitic effects. The impact of these parasitic signals on the measured sensor output can then be estimated and adjusted using this replica. Compared to the prior method [16,18,19], the current approach essentially helps to isolate the genuine signal of interest from the unwanted effects of parasitic variables, resulting in more practical, precise, and dependable measurement (or sensing) performance without the need for modifications to the resonator device used.

This study evaluates the effectiveness of the replica subtraction method and how the ideal resonance spectral shape of the electrothermal-piezoresistive-cantilever sensor is preserved when subjected to a cigarette smoke environment. Here, an Analog Discovery 2 oscillator is controlled in a sweeper application, utilizing phase error in the PLL-based case. Fast Fourier Transform (FFT) analysis is also utilized in computing the amplitude and phase spectra of the resonant sensor signal. Moreover, baseline equations are used to aid in maintaining an optimized frequency response during resonance tracking. The frequency-sweeper application also characterizes spectral responses using the proposed subtraction process for optimizing spectral responses. The hypothesis suggests that the PLL, guided by the phase error, effectively maintains resonance frequencies amidst repeated exposure to cigarette smoke, implying precise and reliable frequency tracking capabilities.

## 2. Electrothermal-Piezoresistive-Cantilever Sensor

An electrothermal-piezoresistive-cantilever sensor was developed to detect numerous analyte targets, such as particles, gas, and biomolecules. The configuration of the electrothermal-piezoresistive-cantilever sensor is depicted in Figure 1, showcasing the integration of the piezoresistive U-shaped Wheatstone bridge and the L-shaped heating resistor within the cantilever beam structure. The resistive component embedded in its cantilever beam makes it a self-actuated and self-sensing resonance sensor. The main electrical components are a *p*-doped piezoresistive Wheatstone bridge (for sensing) and a *p*-doped heating resistor (for actuation). The mechanical actuation that converts thermal power through Joule heating [20] is created through the heating resistor. The thermal power (*P_ac_*) is generated by supplying the heating resistor (*R_h_*) with an actuation signal consisting of a sine-wave ($V_{ac}\times sin\omega t$) superimposed onto a constant voltage (*V_dc_*). The dynamic amplitude *V_ac_* leads to two excited frequencies, i.e., *ω* and 2*ω*. The component *V_dc_* creates a constant stress on the cantilever beam. If *V_dc_* >> *V_ac_*, the cantilever beam is driven into vibration at *ω*, as the frequency component 2*ω* is negligible. Here, the fundamental frequency of the output signal has the same frequency as the excitation signal. The generated force is subsequently detected by the Wheatstone bridge structure and converted into an electrical signal.

However, a direct thermal coupling from the heating resistor to the Wheatstone creates a parasitic crosstalk effect on the sensor output. The close spacing between the two parts leads to a transfer/coupling of parasitic signals, i.e., from actuation (heating resistor) to the sensing (Wheatstone bridge) parts [21,22], which is generally called parasitic feedthrough. As illustrated by the electrical equivalent circuit in Figure 1a, the alternating Ohmic loss (*P_ac_* = 2*V_dc_V_ac_*/*R_h_*, corresponding to an equivalent thermal current source) causes temperature fluctuations (*T_ac_*, corresponding to an equivalent thermal voltage signal) in the heating resistor, formulated as:(1)$$T_{ac}=P_{ac}Z_{th}=\frac{2V_{dc}V_{ac}R_{th}}{R_{h}(1+R_{th}C_{th}s)}$$
where, *Z_th_* is a thermal impedance, *s* = *jω* is the Laplace transform variable, and *R_th_* and *C_th_* are the heating resistor’s effective thermal resistance and capacitance, respectively, which define the resonance frequency *ω*_0_ = 1/*R_th_C_th_*. Essentially, *T_ac_* leads to nonlinear electromagnetic waves interacting with the silicon substrate, generating a “continuum background” coupled to the Wheatstone bridge output signal.

The parasitic crosstalk effects are significant obstacles to characterizing MEMS resonators via electrical measurements. This condition presents an unavoidable parasitic element, resulting in a non-ideal spectral of the amplitude and phase response (delineated in Figure 1b), called an asymmetric spectral shape (black line) and a reversing phase (red line), respectively. The asymmetric spectra are always accompanied by the reversing phase response, which has ambiguous characteristics and is detrimental to real-time operation with a phase-locked loop [23,24]. Therefore, optimizing both the amplitude and the phase spectra is essential. Here, we propose a parasitic-effect elimination method by subtracting a replication of the spectral resonance baseline (to be de-embedded) from the measured signal.

## 3. Resonant Frequency Transfer Function Analysis

Understanding the spectral responses of systems is pivotal in various fields, from engineering to the biological sciences. Frequently, these responses exhibit asymmetric characteristics and reversing phase behavior, necessitating precise mathematical modeling. One such model widely utilized for this purpose is the transfer function *TF*(*s*), as outlined by Equation (2) [25]:
(2)$$TF(s)=\frac{s^{2}+s+q_{1}}{s^{2}+s+q_{2}}$$
where, *q*_1/2_ are constants that contribute as asymmetry factors. Here, *q*_1_ ≠ *q*_2_. To construct an asymmetrical line shape and reversing phase response, the *q*_1_ value (as the numerator) should have a slightly different value than the denominator, *q*_2_. In addition, the *q* value also contributes to determining the resonance condition; the peak of the curve is shifted with *q* variation. Figure 2 illustrates the Bode diagrams of *TF*_1_(*s*), *TF*_2_(*s*), and *TF*_3_(*s*) related to Equations (3)–(5), respectively. Herein, assuming Δ*q* = *q*_2_ − *q*_1_ ≈ 1, the bigger the *q*_1/2_ values, the more the resonant frequency increases. Contrarily though, with increasing *q*_1/2_ values, the amplitude simultaneously decreases toward the same baseline. Suppose the transfer function *TF*_2_(*s*) is subtracted from *TF*_1_(*s*), the result is *RTF*_1_(*s*) (see Equation (6)). Likewise, the operation of *TF*_1_(*s*) − *TF*_3_(*s*) yields *RTF*_2_(*s*), as denoted in Equation (7). Bode diagrams corresponding to *RTF*_1_(*s*) and *RTF*_2_(*s*) are plotted in Figure 3. By subtracting *TF*(*s*)_2_ or *TF*(*s*)_3_ from *TF*(*s*)_1_, near-Lorentzian magnitudes and monotonic phase curves are revealed around the resonance state of the *TF*_1_(*s*), i.e., at ~3.2 rad/s.
(3)$$TF_{1}(s)=\frac{s^{2}+s+9}{s^{2}+s+10}$$
(4)$$TF_{2}(s)=\frac{s^{2}+s+99}{s^{2}+s+100}$$
(5)$$TF_{3}(s)=\frac{s^{2}+s+1000}{s^{2}+s+1001}$$
(6)$$RTF_{1}(s)=\frac{90}{s^{4}+2s^{3}+111s^{2}+110s+1000}$$
(7)$$RTF_{2}(s)=\frac{991}{s^{4}+2s^{3}+1012s^{2}+1011s+10010}$$

These observations underscore the importance of understanding the transfer function’s behavior in capturing spectral responses, particularly in systems with asymmetric and reversing phase characteristics. Here, we delve deeper into the analysis and implications of these responses, elucidating the underlying mechanisms governing spectral behavior.

## 4. Spectral Response Optimization Method and Experimental Evaluation

### 4.1. A Replica of the Baseline Spectral for Signal Optimization

An asymmetric line-shape spectrum yielded by an electrothermal piezoresistive resonant sensor can be related to Fano resonance phenomena. A Fano resonance is supposed to be created from a superposition of a constant amplitude signal with a Lorentzian-line-shaped signal [26]. According to the subtraction operation of the transfer functions above, it can be determined that subtracting a constant spectral response, either for magnitude or phase, from the output of the resonant sensor reveal a Lorentzian amplitude shape and a monotonic phase response. Therefore, a linear equation is proposed as the replica’s governing rule for mirroring the baseline spectral characteristics of the sensor, whether in terms of amplitude or phase, serving as the subtractor component. In the mathematical model (*TF*) above, this replica is represented as the spectral part of *TF*_2_(*s*) and *TF*_3_(*s*) around the resonance state of *TF*_1_(*s*). These subtraction components are subsequently involved in an equation system adopted from phasor subtraction and formulated as follows:(8)$$A_{sen}∠\theta_{sen};A_{sub}∠\theta_{sub}$$
(9)$$x_{1}=A_{sen}\cos\theta_{sen};y_{1}=A_{sen}\sin\theta_{sen}$$
(10)$$x_{2}=A_{sub}\cos\theta_{sub};y_{2}=A_{sub}\sin\theta_{sub}$$
(11)$$A_{sub}=m_{A}f+C_{A};\theta_{sub}=m_{\theta }f+C_{\theta }$$
(12)$$\Delta x=x_{1}-x_{2};\Delta y=y_{1}-y_{2}$$
(13)$$R=\sqrt{\Delta x^{2}+\Delta y^{2}}$$
(14)ϕ=arctan2(Δy,Δx)
where, *A_sen_*, *θ_sen_*, *A_sub_*, *θ_sub_*, *m_A_*, *m_θ_*, *f*, and *C* are signal amplitude, signal phase, subtraction amplitude, subtraction phase, slope or gradient of amplitude (*A*) and phase (*θ*), working frequency, and constant offset, respectively. Signal amplitude and signal phase refer to the magnitude of the voltage signal output by the sensor and the phase difference between the actuation signal and the sensing signal, respectively. In the same context, the same applies to the subtraction amplitude and the subtraction phase, which serve as replicas of the sensor baseline. Here, arctan2(Δ*y*,Δ*x*) returns the angle *φ* between the ray to the point (*x*,*y*) and the positive *x*-axis, confined to (−π, π), as illustrated in Figure 4. This approach accomplishes the previous work that manually introduced constant values [27] as amplitude and phase subtraction components. Subtraction components in a linear equation are expected to be more adaptive, following the sensor spectral dynamic. Eventually, they can better maintain the optimized amplitude (*R*) and phase (*φ*) spectra.

A pocket-sized system based on a low-cost Analog Discovery 2^TM^ (AD2), Digilent Inc. (Pullman, WA, USA) micro controller is considered the primary device to actuate the resonator and acquire and process the sensor signal. The AD2 device is built on a Field Programmable Gate Array (FPGA) that avoids problems concerning sharing timing on microcontroller-based systems [28]. The AD2 can be controlled via application program interfaces (APIs) [29]. The latter allows the instrument/device to be controlled by an external program (such as LabVIEW); hence, it can process the data independently. The circuitry system shown in Figure 5 provides an actuation signal port (WG) linked directly to the cantilever resonator’s heating resistor and fed back to scope channel 2 (CH2). The device subsequently reads the sensor output through channel 1 (CH1). Hence, in this way, the acquired data is streamed to LabVIEW for subsequent processing with the optimizer equations. Using LabVIEW, a frequency-sweeper and phase-locked loop (PLL)-based application system was developed to identify and track the sensor’s resonant frequency. In the case of the frequency-sweeper application, a program application essentially controls the internal AD2 oscillator to complete a set of ‘start and stop’ frequencies and governs the oscillator by a piece of phase error in the PLL case. To calculate the signal amplitude (*A_sen_*) and signal phase spectrals (*θ_sen_*), we implement a Fast Fourier transform (FFT) value of the signal acquired from the cantilever sensor. Furthermore, the baseline equations are formulated to obtain suitable replica spectral amplitude and phase values as subtraction components. According to Equation (11), the amplitude or phase of subtraction components behave as frequency functions. Subsequently, these baseline equations are subtracted from the amplitude and phase signals measured by the sensors, in accordance with Equation (12). The optimized spectra are then meticulously computed utilizing the formulations in Equations (13) and (14). An optimized frequency response is expected to be maintained during resonance tracking using the proposed replica spectral method.

As an assessment, the baseline replica subtraction approach was evaluated to observe how effectively it can maintain the optimum condition of the cantilever’s resonance spectral shape during varying analyte-exposure conditions. The assessment was conducted by putting the resonant sensor under a cigarette-smoke environment. Under this condition (i.e., a large number of particles), a considerable resonance shifting is expected, since the suspended cigarette smoke particles get attracted to the cantilever beam via electrophoresis/dielectrophoretic processes [30]. The optical microscope photograph in Figure 6 indicates that the humid fractions of cigarette smoke (expected water content and sticky components, such as tar) are attached to the cantilever beam.

Furthermore, spectral responses were recorded by the frequency-sweeper application of the measurement system designed to employ the proposed subtraction process for optimizing spectral responses. Regarding an assessment procedure, the resonance spectral curve of the sensor without smoke exposure is taken as the initial condition. Afterwards, the spectral response was measured after the first, second, and third cigarette smoke exposures. Figure 7 demonstrates the resonance spectral curves of the cantilever amplitude (a) and phase (b) and their respective subtractor curves (i.e., the replica spectral) before and after the smoke exposure.

A non-ideal configuration between the sensor and the replica spectral curves occurring after the first smoke exposure led to “disrupted” optimized amplitude and phase responses (Figure 8) for the following exposure conditions. Changes to the sensor spectral baseline after the first smoke exposure, moving it away from the determined replica spectral curve, were ascribed to a temperature increase from ~26.8 °C (cantilever without smoke) to ~28.6 °C (cantilever at the first smoke exposure). Furthermore, a reversing trend (shown as a disrupted part in the not finally optimized phase response) potentially will disturb frequency locking by the PLL system due to the appearance of ambiguity in the resonance phase.

Since the sensor’s baseline moves to another level after the first smoke exposure, it is necessary to re-adjust another replica spectral curve to fit in an ideal configuration with the “new” sensor’s baseline. The process of re-adjusting the replica spectral consistently produces acceptable deviations in the resonant frequency, observed both before and after re-adjustment. Utilizing the fitting method [24] in both scenarios uncovers a deviation of approximately 3 Hz. Specifically, the resonant frequencies are measured as 261.13 kHz before re-adjustment and 261.16 kHz after re-adjustment, respectively. Figure 9 exhibits an adaptive arrangement of the spectral structure after re-adjusting to the second replica spectral. The ideal placement of the sensor’s baseline and the replica spectrum for amplitude and phase generate a symmetrical amplitude shape (Figure 9a) and monotonic phase transition (Figure 9b). In respect to the optimized response, a sufficiently large Δ*f*_0_ ≈ 3.63 kHz is observed from the initial state to the third smoke exposure. Furthermore, monotonic phase responses are demonstrated, resulting in an almost horizontal shift direction. This is crucial for the validity of resonance locking in a PLL system. Moreover, thanks to the reasonably stable ambient temperature (28.6 ± 0.18 °C) and relative humidity (33.9% ± 3.26%) conditions, one set of replica spectral baseline equations (the subtraction operator) can be applied for the three different conditions (i.e., first, second, and third smoke exposures) without the need for further re-adjustment. Hence, environmental stability is essential for performing the replica spectral baseline method during PLL-based resonance tracking. In practical usage, if the system is subjected to unstable environmental conditions, the employment of a periodic spectral sweep (utilizing the frequency-sweeper application) is essential. This process ensures continuous monitoring and maintenance of the sensor and subtraction spectra, keeping them in their optimized configurations.

### 4.2. PLL-Based Cantilever-Resonance Tracking

A phase-locked-loop (PLL)-based resonance tracking of the electrothermal cantilever needs a monotonic phase response excited by the resonance sensor. With a monotonic phase response, the PLL can lock the resonance phase (*φ*_0_) at a specific frequency value, subsequently interpreted as the resonant frequency (*f*_0_). PLL-based resonance tracking consists of three main elements, i.e., a phase detector, a controller, and a feedback system, formulated by a LabVIEW-based software code. The phase detector measures the phase between the actuation signal and the sensor’s output. This is achieved by measuring the time difference between the zero crossings of both signals. This time difference is proportional to the phase. Afterward, the controller uses the phase error to track the resonant frequency. As shown in Figure 10, the PLL locks the resonant frequency properly within three different cigarette smoke exposures. The resonant frequency decrement due to the cigarette-smoke-shoot collection is determined over a 10-min sampling period. A tracking rate of 0.5 Hz/s and a phase error of 0.0176° is exhibited by setting a gain of 1 in the AD2-PLL application, which leads to a standard error of 1.6 mHz for the stable-locked frequency. A comparison between the tracked resonant frequency and the identified resonant frequency of the optimized amplitude spectral obtained through the sweeper application is summarized in Table 1. Overall, the PLL can quite precisely lock or track the resonant frequency. Referring to the frequency-sweeper result, the average deviation of the tracked frequency under the initial state and after three instances of smoke exposure is about 37.5 Hz. It leads to a quite small error measurement of ~14%. To enhance the precision of the locking process, several strategies may be employed, including parameter adjustment within the PLL control system, optimization of the PLL control algorithm, and attenuation of noise through the application of electronic filtering.

Furthermore, to transpose the resonance shift to the smoke particle-mass-concentration regime, we employ a calibration factor of 500 (µg·min)/(m^3^·Hz) [9]. The full-square-red line in Figure 10 indicates that the appearance of three instances of smoke exposure can be visualized. Particle mass concentrations of ~24 mg/m^3^, ~31 mg/m^3^, and ~47 mg/m^3^ are the highest concentrations of the first, second, and third of cigarette smoke exposure, respectively. Overall, the average deviation of the mass concentration is identified at ~6.5 mg/m^3^, in which the higher deviation values mainly occurred when the smoke concentration was increased. The relative humidity and temperature also increase when the smoke concentration rises. Hence, the fluctuations in relative humidity and temperature expected during smoke exposure, i.e., 10.4% and 0.5 °C, respectively, are fathomedto have a prominent effect on the sensor’s accuracy. Additionally, if we consider the deviation of 37.5 Hz between the frequency-sweeper and PLL in the resonance state determination, this deviation corresponds to a mass concentration of approximately 1.9 mg/m^3^.

## 5. Conclusions

The presented parasitic de-embedding scheme works out properly with the reversing phase responses caused by the crosstalk between the heating resistor and the piezoresistor of an electrothermal-piezoresistive-cantilever sensor. This scheme has become a practical approach for mitigating the asymmetric spectral line shape generated by cantilever sensors. It provides a kind of adaptive subtraction of a replica of the crosstalk effect that yields a stable monotonic phase response and demonstrates good frequency tracking. Moreover, since the optimization process is independent of sensor properties, this approach is also flexible and valid for other resonant sensors (e.g., piezoelectric-/capacitive-based sensors). In unstable environmental conditions, a periodic spectral sweep (using the frequency-sweeper application) ensures that the sensor and the subtraction spectra are maintained in their optimized configurations.

## Figures and Tables

**Figure 1 sensors-24-02318-f001:**
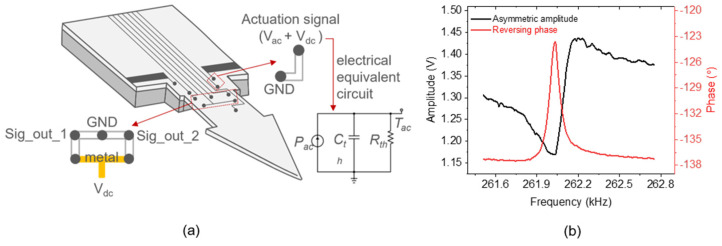
(**a**) Illustration of an electrothermal-piezoresistive-cantilever sensor depicting its U-shaped sensing component and L-shape heating resistor, accompanied by a schematic diagram illustrating the electrical equivalent circuit diagram of the heating resistor system. Passing an AC-current (*i_ac_* = *P_ac_*/(2*V_dc_*)) through the heating resistor results in a fluctuating temperature change (*T_ac_*). (**b**) Non-ideal frequency response of an electrothermal-piezoresistive-cantilever sensor showing an asymmetric amplitude (black) and reversing phase response (red).

**Figure 2 sensors-24-02318-f002:**
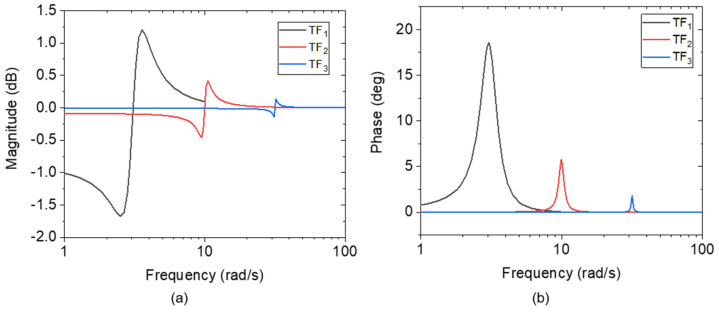
Bode diagrams of *TF*_1_ (black), *TF*_2_ (red), and *TF*_3_ (blue) depicting a decrement of (**a**) magnitude/amplitude and (**b**) phase responses with increasing *q*_1/2_ values in line with Equation (1), and simultaneous shifting of resonance amplitude toward the same baseline.

**Figure 3 sensors-24-02318-f003:**
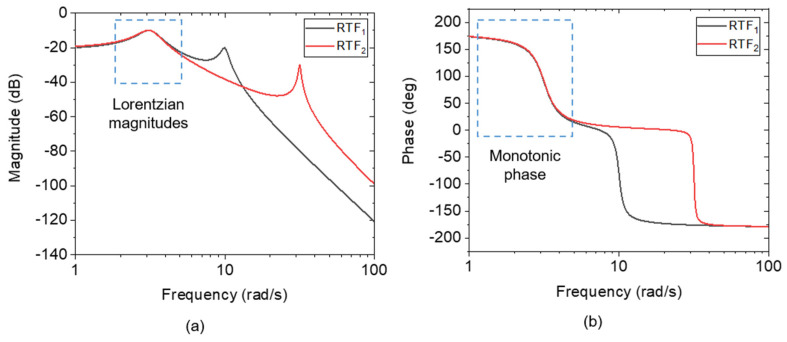
Bode diagram of *RTF*_1_ (black) and *RTF*_2_ (red) performing a (**a**) Lorentzian amplitude, and (**b**) a monotonic phase response at the resonance state of *TF*_1_.

**Figure 4 sensors-24-02318-f004:**
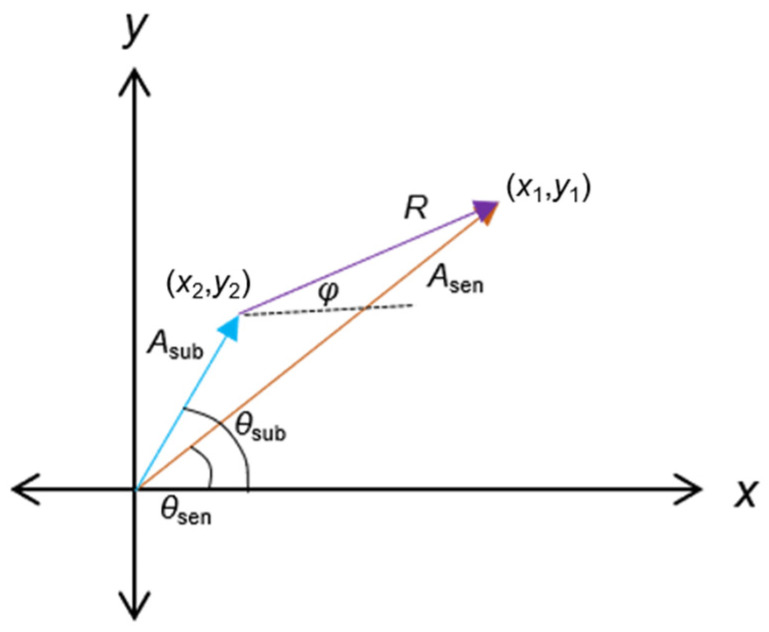
Phasor subtraction of sensor and replica of the baseline components (amplitude *A_sub_*, phase *θ_sub_*) resulting in the optimized amplitude *R* and phase *φ* direction.

**Figure 5 sensors-24-02318-f005:**
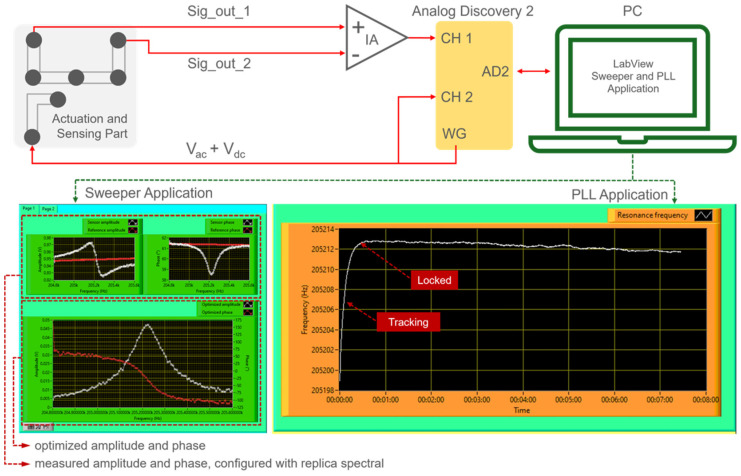
AD2−electrothermal-piezoresistive-cantilever circuitry system connected to LabVIEW as the user interface for sweeper and PLL applications.

**Figure 6 sensors-24-02318-f006:**
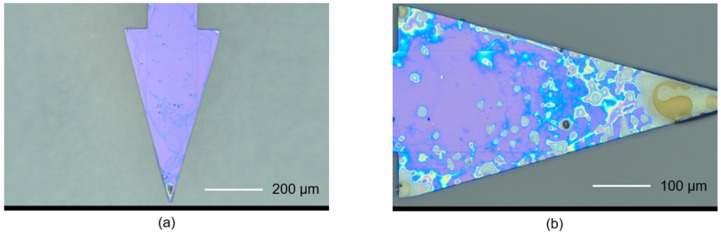
Optical microscope graph showing a triangular-type cantilever beam before (**a**), and after (**b**) cigarette smoke exposure.

**Figure 7 sensors-24-02318-f007:**
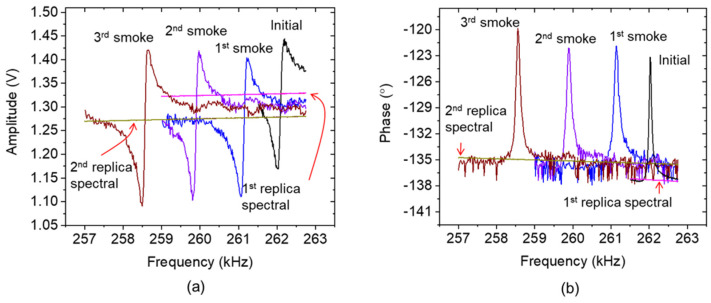
Arrangement of the measured original electrothermal-piezoresistive-cantilever responses with (**a**) asymmetric amplitude spectral curve accompanied by (**b**) a reversing phase response and the replica of the baseline spectral for spectral optimization under different cigarette smoke conditions.

**Figure 8 sensors-24-02318-f008:**
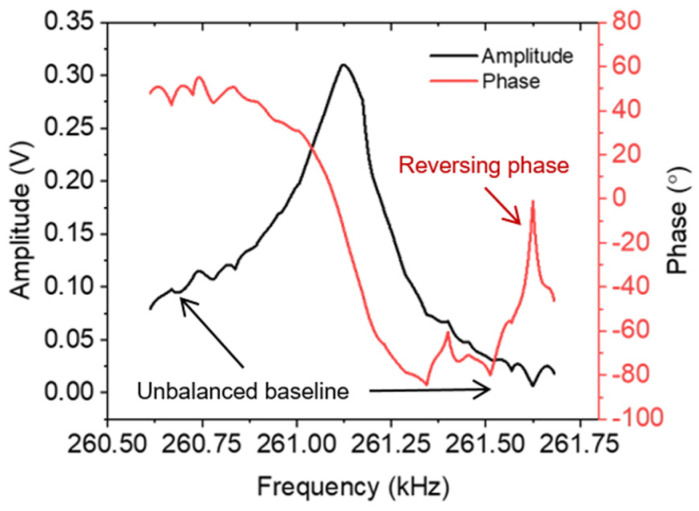
Non-ideal arrangement between electrothermal-piezoresistive-cantilever response and a baseline replica disrupting an optimized spectral, leading to an imbalance in the amplitude baseline and a reversal in the phase curve.

**Figure 9 sensors-24-02318-f009:**
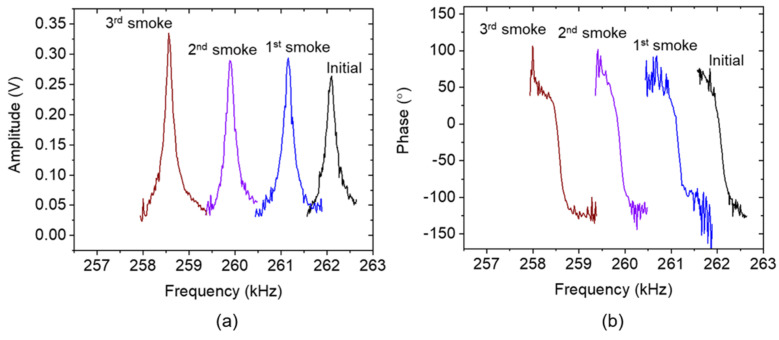
Re-adjustment of the imitated baseline results in steady (**a**) amplitude and (**b**) phase spectrals with a constant resonance phase of −20°.

**Figure 10 sensors-24-02318-f010:**
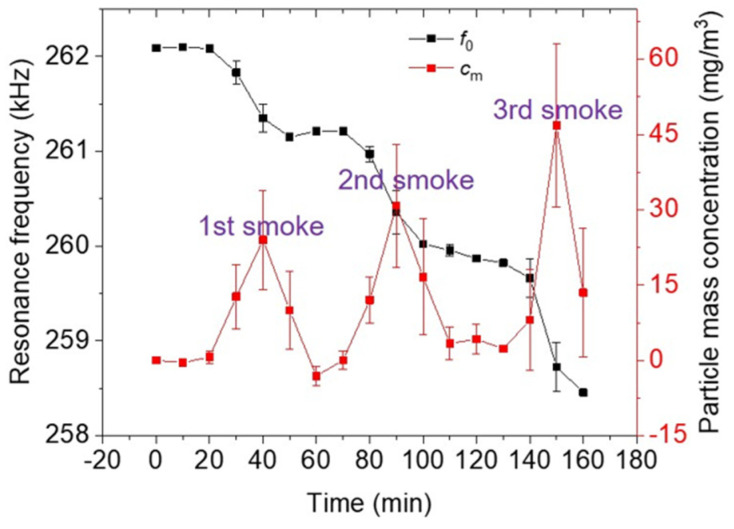
Resonance locking process under three different cigarette smoke exposure conditions.

**Table 1 sensors-24-02318-t001:** Comparison of the tracked resonant frequency versus frequency-sweeper-identified resonant frequency.

Smoke Exposure Condition	Sweeper (kHz)	PLL Tracking (kHz)	Deviation (Hz)
Initial	262.10	262.09	10
1st smoke	261.16	261.19	30
2nd smoke	259.90	259.88	20
3rd smoke	258.55	258.46	90

## Data Availability

Data are unavailable due to privacy restrictions.

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
