# Peer review of "Replicating Spectral Baseline for Unambiguous Frequency Locking in Resonant Sensors"

_sensors, 2024, doi:10.3390/s24072318_

Round 1

Reviewer 1 Report

Comments and Suggestions for Authors

1. The authors should give a microscope photograph of the sensors with heating and sensing parts marked in Fig.5

2. In line 220. it should be (a) amplitude and (b) phase.

3. The sensor’s baseline moves to another level after each smoke exposure, is it possible that the re-adjusting sensor's baseline may cause some uncertain issue?

4.How to improve the performance of "average deviation of the tracked frequency is about 37.5 Hz" in line 286-287.

5. In fig.9 , a detailed illustration of sweeper and PLL application system should be provided. 

Author Response

The author's response is provided in the attachment.

Reviewer 2 Report

Comments and Suggestions for Authors

The paper is devoted to increasing the sensing performance of the cantilever-based sensor with thermal actuation and piezoresistive readout. The introduction clearly demonstrates relevance of the work. The proposed method is analyzed theoretically and verified experimentally. The paper is interesting for MEMS society, particularly for people involved in resonant sensing. The manuscript makes positive impression, it is well written and readable. It can be published in Sensors after minor improvement:

Section “2 Materials and Methods” does not describe the sensor design. This information would make the paper more practical and and understandable. In particular, it would be interesting to see the location of the actuation and sensing resistors. Details of the experimental setup for actuation and collection of the output signal are also welcome.

Author Response

(The authors gave the same response as above.)

Reviewer 3 Report

Comments and Suggestions for Authors

My review is very short. I am very disappointed by the manuscript and first wanted to reject it. However, I want to give the authors one more chance. Simple question - how can we discuss, simulate and investigate SOMETHING without explanation of what we are  speaking about? The manuscruipt starts with some brief general notes, then in one paragraph the authors describe (in a very unclear manner) the device under consideration and then immediately start some simulation of this SOMETHING. No, we first are to see the normal introduction, explaining the device architecture and implementation, special features of its performance etc. This introduction has to be accompanied by the drawing or 3D sketch, by structural scheme and by the normal electric scheme (not the simple picture from the textbook, shown as the Fig.1). Then the problem to be analysed in the paper and solved has to be explained and only then the authors are to explain their original approach and its sense and novelty.

Formulae, mathematical expressions and all parameters and variables are also to be properly introduced and explained. The authors are also to check their scientific English. The language of the manuscript is pretty good, but the terms often seem to be incorrect - for instance, in the page 5 the authors use the terms like "sensor amplitude" or "sensor phase" etc.

The authors use the now-popular term (better to say - a kind of mem) - a "digital twin". Various people use it in a very different sense, so the authors are, at least, to introduce this term in a proper formal way. More specifically, the procedure of the "digital twin subtraction" has also to be explained and properly introduced (with references).

Comments on the Quality of English Language

English is pretty good, but the terms are to be checked - some of them look like a slang.

Author Response

(The authors gave the same response as above.)

Round 2

Reviewer 1 Report

Comments and Suggestions for Authors

The authors have improved the manuscript and it is ready to accept. 

Reviewer 3 Report

Comments and Suggestions for Authors

I am satisfied by corrections and addend. The paper may be accepted now.